# Association of Ramadan Participation with Psychological Parameters: A Cross-Sectional Study during the COVID-19 Pandemic in Iran

**DOI:** 10.3390/jcm11092346

**Published:** 2022-04-22

**Authors:** Hesam Addin Akbari, Mohammad Yoosefi, Maryam Pourabbas, Katja Weiss, Beat Knechtle, Rodrigo Luiz Vancini, Georgia Trakada, Helmi Ben Saad, Carl J. Lavie, Amine Ghram

**Affiliations:** 1Department of Exercise Physiology, Faculty of Physical Education and Sport Sciences, University of Tehran, Tehran 1415563117, Iran; hesameakbary@ut.ac.ir (H.A.A.); mohammad.yoosefi@ut.ac.ir (M.Y.); pourabbas.maryam@yahoo.com (M.P.); 2Medbase St. Gallen Am Vadianplatz, 9100 St. Gallen, Switzerland; katja@weiss.co.com (K.W.); beat.knechtle@hispeed.ch (B.K.); 3Institute of Primary Care, University of Zurich, 8091 Zurich, Switzerland; 4Center for Physical Education and Sports, Federal University of Espírito Santo, Vitória 29075-910, Brazil; rodrigoluizvancini@gmail.com; 5Department of Clinical Therapeutics, National and Kapodistrian University of Athens, School of Medicine, 115 28 Athens, Greece; gtrakada@hotmail.com; 6Research Laboratory “Heart Failure, LR12SP09”, Faculty of Medicine of Sousse, Hospital Farhat HACHED of Sousse, University of Sousse, Sousse 4054, Tunisia; helmi.bensaad@rns.tn; 7Healthy Living for Pandemic Event Protection (HL—PIVOT) Network, Chicago, IL 60612, USA; clavie@ochsner.org; 8Department of Cardiovascular Diseases, John Ochsner Heart and Vascular Institute, Ochsner Clinical School, The University of Queensland School of Medicine, New Orleans, LA 70121, USA

**Keywords:** exercise training, feasting, holy month, lockdown, mental health, SARS-CoV-2, sleep-quality

## Abstract

COVID-19 restrictions are associated with poor physical-activity (PA). Less is known about the relationship between the combination of these restrictions with Ramadan intermittent fasting (RIF), PA, mental health, and sleep-quality. The present study aimed to evaluate whether COVID-19 restrictions and RIF during the fourth wave of the COVID-19 pandemic in Iran are associated with poor PA, anxiety, well-being, and sleep-quality outcomes. A total of 510 individuals participated in an online questionnaire that was disseminated to adults (≥18 years) residing in Iran from 13 May 2021 to 16 May 2021 (~3 days), just after the end of Ramadan 2021. PA behavior (Godin-Shephard Leisure-Time Exercise Questionnaire), anxiety (General Anxiety Disorder-7), well-being (Mental Health Continuum-Short Form), and sleep-quality (Pittsburgh Sleep Quality Index). Of 510 individuals included in the study (331 female (64.9%); mean ± SD, 31 ± 12 years), 172 (33.7%) reported less PA during the Ramadan 2021. PA was associated with better well-being and sleep-quality outcomes. Regardless of PA, participants who fasted for all of Ramadan had less anxiety and better well-being outcomes than those who fasted part of Ramadan or did not fast at all. However, the fasting part of Ramadan decreased the sleep-quality of active participants. The Ramadan 2021 was associated with poor PA, well-being, and sleep-quality of Iranians. However, PA was associated with better well-being and sleep-quality outcomes, and those who fasted all Ramadan had better anxiety and well-being outcomes. Therefore, PA during Ramadan might be an essential and scalable mental health resilience builder during COVID-19 restrictions which should be encouraged.

## 1. Introduction

Iran is one of the most affected countries by the coronavirus disease (COVID-19). The COVID-19 fourth wave in Iran started at the end of March 2021 [1] and lasted longer than previous waves [2]. The Iranian health authorities imposed restrictions, such as isolation and lockdown, to reduce the prevalence of COVID-19 [3]. These preventive health measures have been shown to have negative consequences, such as decreased physical-activity (PA) levels [4,5,6,7,8,9,10,11], sleep-quality [3,4,5,6,7,8,9,10,11,12], and increased depression and anxiety [3,13,14,15,16,17]. Importantly, the holy month of Ramadan 2021 (the ninth month in the Islamic calendar, and one of five pillars of Islam) coincided with the COVID-19 pandemic, which can pose several challenges for Muslim people, given the persistence of COVID-19 outbreaks, the appearance of new coronavirus mutations, and the mass vaccination campaign [4]. In Iran, Ramadan 2021 began on 12 April, lasted 30 days, and fasting duration typically ranged from 14 to 16 h.

Iranian Muslims celebrated Ramadan under the shadow of the fourth outbreak of COVID-19. Ramadan fasting is a type of intermittent fasting practiced by millions of healthy adult Muslims [18]. Ramadan intermittent fasting (RIF) is obligatory for healthy adult Muslims. However, some people are exempt from fasting, such as those who are susceptible to COVID-19 by fasting, or with COVID-19, or with higher risk for adverse health events, as well as older people who are physically weak [4,5,6,7,8,9,10,11,12,13,14,15,16,17,18,19].

A previous study showed the health benefits and potential health concerns of fasting [20]. RIF represents a new concern for Muslim people for dealing with COVID-19 [4,5,6,7,8,9,10,11,12,13,14,15,16,17,18,19,20,21]. On the one hand, previous studies have shown an alteration of dietary habits, the amount and pattern of PA, and the duration of sleep during Ramadan [22]. On the other hand, fasting, with psychological and spiritual relaxation, decreases stress and anxiety [20]. The lack of relevant studies and its association with changes in sleeping activity patterns [23] and circadian rhythms of hormones [18] raised growing interest concerning the impacts of RIF on Muslim psychological heath in the era of the COVID-19 pandemic. During COVID-19 restrictions, RIF may reduce the PA level and induce a reversal of positive physiological and psychological adaptations associated with PA [4]. Thus, Ramadan 2022 starts on 2 April and will pose an additional concern for Muslim people with the persistence of the COVID-19 outbreak.

The study’s research question was whether there is an association between RIF and PA/psychological parameters, such as PA levels, anxiety, well-being, and sleep-quality patterns in the era of the COVID-19 pandemic in Iran.

## 2. Population and Methods

The present study is part of a project involving two parts. The first, which was recently published [3], aimed at investigating the impacts of COVID-19 restrictions on PA behavior, well-being, anxiety, and sleep-quality in the Iranian population. The second part is the objective of this study. For the above reason, a large part of the methodology of this study has already been previously described [3].

### 2.1. Participants

To participate in this survey study, participants had to be Iran residents aged ≥18 years. Participants were recruited through snowball sampling using social media. Participants did not have to state their names or contact information, and they could stop participating in the study at any moment without saving their answers and personal data. The survey was designed to make the answers to all its clauses mandatory. By pressing the submit button participants consented to participate in the study and only then were the answers saved to the database. This study was conducted following the Declaration of Helsinki and received approval from the Institute of Physical Education and Sports Sciences (Tehran, Iran) Human Subject Committee (IR.SSRC.REC.1400.034).

The sample size was calculated as follows [24]: *N* = (*Z*_1−*α*/2_)^2^ (*p*) (1 − *p*)/(*d*)^2^, where “*N*” indicates the number of individuals needed; “*Z*_1 − *α/2*_”, standard normal variate at 5% type one error (1.96); “*p*”, expected proportion in the physically active population (0.48) based on a previous study [3]; “*d*”, absolute error or precision (0.05). The sample size was, therefore, 384 participants.

### 2.2. Measures

Participants completed the Persian version of an online survey created using the PORSA (https://porsa.irandoc.ac.ir/) questionnaire maker from 13 May 2021 to 16 May 2021 (~3 days), just after the end of Ramadan 2021 (accessed on 5 May 2021). The survey comprised four standardized questionnaires regarding PA behavior, anxiety, well-being, and sleep-quality.

#### 2.2.1. Demographic Information

Demographic characteristics included age, sex, height, weight, marital status, schooling-level, occupational status before and during Ramadan, living environment and location. Participants were asked to determine the fasting status during Ramadan, the reason for not fasting among people who did not fast or fasted just some days during Ramadan, religion, ethnicity, vaccination status against COVID-19, and history of getting COVID-19 before and during Ramadan.

#### 2.2.2. Primary Outcomes and Measures: PA Behavior

We used the Persian version of the Godin-Shephard Leisure-Time Exercise Questionnaire (GSLTEQ) to assess individual PA behavior over the previous week [3,25], which has been shown to have acceptable reliability in Iranian populations (reliability coefficient = 0.79) [25]. We divided individuals according to their GSLTEQ score: ≥24 (active), ≤23 (inactive) [26]. Similar to a previous study which assessed how the preventive measures impacted PA behavior and well-being of Canadians [27], the cut off for active was >150 min of moderate–vigorous PA per week while the related cut off for inactive was <149.9 min of moderate–vigorous PA per week [28]. We also asked further questions to assess current PA behavior: (i) whether PA had changed since Ramadan, (ii) the type of PA most frequently exercised during Ramadan, (iii) whether the type of PA had changed since Ramadan, (iv) location of PA during Ramadan, and (v) whether the location had changed during Ramadan. 

#### 2.2.3. Secondary Outcomes and Measures: Anxiety, Well-Being, and Sleep-Quality

We used the Persian version of General Anxiety Disorder-7 (GAD-7) to assess individual anxiety symptoms [3,29], which has been shown to have good reliability for Iranians (Cronbach’s alpha = 0.876) [29]. The final number obtained from GAD-7 divides participants into no anxiety (0–4), mild anxiety (5–9), moderate anxiety (10–14), and severe anxiety (15–21) levels [30]. 

We used the Persian version of the Mental Health Continuum-Short Form (MHC-SF) to assess emotional, psychological, social, and overall well-being [3,31]. The MHC-SF has been previously used with Iranians and found to have excellent reliability for emotional, social, and psychological subscales and total score on the MHC-SF (Cronbach’s alpha = 0.84, 0.88, 0.88, and 0.92, respectively) [31]. The resulting total score of MHC-SF can range from 0 to 70. A higher score means a higher level of well-being [32]. 

We used the Persian version of the Pittsburgh Sleep Quality Index (PSQI) to assess an individual’s sleep-quality over the past month [3,33]. The PSQI has been previously used with Iranian populations and found to have good reliability (Cronbach’s alpha = 0.81) [33]. The global PSQI score ranges from 0 to 21, with higher scores indicating more severe sleep disorders. The global PSQI scores >5 and ≤5 indicate poor and good sleep qualities, respectively [34]. 

### 2.3. Statistical Analysis

Data analysis was completed using the Statistical Package for Social Sciences (SPSS-26.0 software). Demographic characteristics were categorized by sex and summarized using descriptive statistics. Independent *t*-tests and chi-square tests were utilized to compare demographic differences between men and women. To assess the association between PA levels and anxiety, well-being, and sleep-quality, the individuals were split into inactive (IPs) and active (APs) participants. Comparative analysis was conducted utilizing independent *t*-tests and chi-square tests. To assess the association of RIF with anxiety, well-being, and sleep-quality outcomes, the IPs and APs were further split into full-fasting, part-fasting, and no-fasting status, and comparative analysis was conducted utilizing chi-square and analysis of variance (ANOVA) tests. Bonferroni’s post hoc tests were used to identify the significance of pairwise comparison of the mean values among subgroups. Alpha was set at <0.05.

## 3. Results

A total of 525 individuals completed the survey. There were fifteen participants who provided irrelevant information, and their data were excluded from final analysis. Thus, the survey response rate was 97%, and data of 510 individuals were considered for the analysis. The study population comprised 179 men (mean ± SD, 31 ± 13 years; 176 ± 7 cm; 79 ± 14 kg) and 331 women (mean ± SD, 30 ± 12 years; 163 ± 6 cm; 63 ± 13 kg).

Table 1 lists individual demographics categorized by sex. Chi-square tests revealed that most of the participants were women (64.9%), single (62.0%), and had a bachelor’s degree (37.1%). While 35.1% of participants reported they were unemployed, 80.0% indicated they had experienced no change in their employment status because of Ramadan. A total of 91.4% and 37.1% of participants lived in urban areas and central provinces, respectively. There were 66.3% of participants who declared they fasted all or part of Ramadan. The reasons for not fasting were illness or undisclosed personal reasons in 56% of cases. Some people (3.7%) refused to fast for fear of getting COVID-19. The most participated ethnic group was Fars ethnic (57.6%). A total of 89.6% of participants were Shia Muslims, and 6.9% were Sunni Muslims. There were 96.5% of participants who were not vaccinated against COVID-19. A total of 30.8% and 4.1% of participants experienced COVID-19 before and during Ramadan, respectively.

### 3.1. Primary Outcomes: PA Behavior

A total of 68.8% of IPs and 63.2% of APs fasted all or part of Ramadan. Table 2 summarizes the PA behavior of APs and IPs during Ramadan. Compared to IPs, APs spent more time on mild (81%, 95% confidence interval (CI) −0.58 to −0.09; *p* = 0.007), moderate (159%, 95% CI −0.8 to −0.4; *p* < 0.001), and strenuous (181%, 95% CI −0.9 to −0.3; *p* < 0.001) PA, and had higher GSLTEQ scores (180%, 95% CI −1.7 to −0.7; *p* < 0.001). A total of 42.1% of APs became even less active, while only 27.0% of IPs became less active. Comparatively, 2.5% of IPs became more active, while 7.9% of APs became more active during Ramadan. There were 23.4% of IPs and 42.1% of APs that maintained their PA duration during Ramadan. Most of the IPs had no activity at all (52.8%), while the most commonly reported activity for the APs was walking (50.0%). Most APs hardly changed their PA type during Ramadan (80.7%). The PA location of 40.4% of participants was outdoors, and most of them did not change their PA location during Ramadan (49.8%).

### 3.2. Secondary Outcomes:Anxiety, *Well-Being**, and* Sleep-Quality

Figure 1 shows the frequencies of good and poor sleep-quality in APs and IPs. Table 3 summarizes the anxiety and well-being scores between APs and IPs. APs had significantly better emotional (7%, 95% CI −0.015 to −0.002; *p* < 0.014), psychological (11%, 95% CI −0.033 to −0.009; *p* < 0.001), social (13%, 95% CI −0.04 to −0.01; *p* < 0.001), and overall well-being (11%, 95% CI −0.08 to −0.03; *p* < 0.001) outcomes than IPs.

Table 4 summarizes the anxiety and well-being scores between APs and IPs according to their fasting status. ANOVA and Bonferroni’s post hoc tests demonstrated that the anxiety (38%, 95% CI −0.037 to −0.007; *p* = 0.002), emotional (15%, 95% CI 0.004 to 0.030; *p* = 0.006), psychological (20%, 95% CI 0.01 to 0.06; *p* = 0.001), and overall well-being (14%, 95% CI 0.01 to 0.12; *p* = 0.008) outcomes of IPs who fasted all the Ramadan were significantly better than those who did not fast at all. Similarly, APs who fasted all of Ramadan had significantly better anxiety (47%, 95% CI −0.04 to −0.01; *p* < 0.001), emotional (14%, 95% CI 0.002 to 0.031; *p* = 0.017), psychological (12%, 95% CI 0.001 to 0.048; *p* = 0.037), and overall well-being (11%, 95% CI 0.007 to 0.116; *p* = 0.021) outcomes than those who did not fast at all.

The sleep-quality of the APs (10%, 95% CI 0.0001 to 0.0110; *p* = 0.046) was significantly better than the IPs (Table 3). ANOVA and Bonferroni’s post hoc tests revealed that the APs who fully fasted had significantly better sleep-quality than those who fasted part of Ramadan (27%, 95% CI −0.027 to −0.004; *p* = 0.008) (Table 4). However, there was a significant association between RIF status and daytime dysfunction in APs, so that just 22.2% of APs who fasted part of Ramadan had no daytime dysfunction.

## 4. Discussion

The present study investigated the effects of COVID-19 restriction on PA behavior, anxiety, well-being, and sleep-quality during Ramadan 2021 in an Iranian population. Our results showed that most participants fasted all or part of Ramadan. The PA duration of a significant portion of individuals decreased during Ramadan. Compared to IPs, APs had better well-being and sleep-quality outcomes. However, anxiety had no association with PA levels. The IPs and APs who fasted all of Ramadan had significantly better well-being and anxiety outcomes than those who did not fast at all. Therefore, there was an association between RIF and PA, anxiety, well-being, and sleep-quality outcomes during the fourth wave of the COVID-19 in Iran. These findings suggest that healthy adults might benefit from performing PA while fasting, which provides psychological support regardless of their PA level. However, these benefits are more significant in physically active individuals.

During the fourth wave of COVID-19 in Iran and despite the recommendation to maintain PA during the lockdown [6,8,35,36] and Ramadan [4,21], our results showed that COVID-19 and RIF had large impacts on PA levels. A total of 23.4% of IPs and 42.1% of APs maintained their PA duration and 2.5% of IPs and 7.9% of APs became more active during Ramadan. The decrease in PA level may have deleterious public health effects [37] by increasing anxiety and stress levels [3,6,13,17,38], as well as the risk of chronic diseases [3,39,40]. Most participants did not change their PA location during Ramadan amid the fourth wave of COVID-19 in Iran. However, individuals who fast are less inclined to exercise during Ramadan, which worsens PA levels during this month [41]. Therefore, changes in PA behavior and other measured outcomes can be attributed to RIF rather than COVID-19 restrictions. However, the critical role of COVID-19 restrictions cannot be denied because a similar study [3] demonstrated a significant association between COVID-19 restrictions and changes in PA, anxiety, well-being, and sleep-quality outcomes in Iran. Our data suggest that APs had better mental health and sleep-quality outcomes than IPs. These data are consistent with the results of previous studies, which demonstrated that PA can improve mental health [42] and sleep-quality [3] outcomes during the pandemic. According to the World Health Organisation recommendations, adults should do at least 150–300 min of moderate-intensity aerobic PA, or at least 75–150 min of vigorous-intensity aerobic PA, or an equivalent combination of them throughout the week, and 2–3 days a week muscle-strengthening activities [43]. While the APs of the study have followed the minimum duration of aerobic PA recommendations, the IPs that do not follow these recommendations are in a clear majority. Muscle-strengthening activities were practiced only by a small percentage of the participants which adds to the concerns. Additionally, most present study participants reported that they decreased their PA duration in Ramadan. A few months before Ramadan 2021, a study demonstrated that approximately half of the 3323 participating Iranians decreased their weekly PA duration during the COVID-19 pandemic, and just 9.1% of them participated in the muscle-strengthening activities [3]. These data show the deterioration of PA in Iran during Ramadan 2021. The decline in PA during Ramadan is worrying, as it has been shown that this decline in PA levels can extend beyond Ramadan [44]. Fasting can reduce the risk of cardiometabolic diseases by affecting the individual’s weight, which can reduce the risk of mortality and morbidity. Since obesity by itself is a risk factor for COVID-19 [45], performing regular PA and fasting may directly reduce obesity and affect the rate of COVID-19 infection, as well as its severity [9,11,36,46].

The present study showed that during the fourth wave of COVID-19 in Iran, individuals who fasted during Ramadan had better well-being and anxiety outcomes than those who did not fast at all, and individuals who performed regular PA while fasting all of Ramadan had better sleep-quality than those who fasted part of Ramadan. These findings indicate that fasting plays an effective role in reducing psychological health problems. The findings in this research are in agreement with the results of Bayani et al. [47] who showed that Ramadan has spiritual benefits and has been associated with positive effects on overall psychological well-being outcomes of those who fast. On the other hand, sedentary behavior and physical inactivity are potential causes of psychological problems and performing regular PA while fasting can be a useful solution to this problem.

Although sleep plays a substantial role in physical and mental health [48], sleep durations and times have been reported to be negatively affected by RIF [49,50,51]. In our study, Ramadan was also associated with reduced sleep-quality, especially in IPs. This finding is in agreement with a previous study which showed an alteration in sleep-quality and physical performance in athletes [52]. Only APs who fasted all of Ramadan had better sleep-quality than those who fasted part of Ramadan. In accordance with previous meta-analyses, athletes and physically active men, who continue to train at least twice/week during Ramadan, experience a reduction in sleep quantity [53,54]. The reduced sleep duration may be explained by the large amount of food consumed at night [23], eating close to bedtime [55], and the increase of night-time social activities [53,56]. Our finding indicates the importance of PA to maintain a good status of sleep-quality, although the prevalence of sleep disorders could be aggravated during the month of Ramadan [21]. According to our findings, performing PA while fasting for all of Ramadan is beneficial for sleep-quality. In this regard, PA and fasting can improve physical and mental health and represent substantial behavior to combat against COVID-19 infection and its severity [4,9,11,21].

The present study demonstrated that individuals who performed partial fasting had poorer sleep quality than individuals who did not fast (non-significant). This result could be explained by the dangers of fasting if it is not done correctly which can increase stress levels and disrupt sleep. Partial fasting could be considered compatible with a non-Ramadan dietary approach. Non-RIF can take different forms such as time-restricted feeding/eating [57], alternate-day fasting [58], and intermittent energy restriction [59]. Although there is compelling evidence of the benefits of intermittent fasting forms on physiological [57,58,59] and psychological parameters [60], it is unknown whether there are psycho-physical differences between fasting all Ramadan and fasting part of Ramadan. Thus, additional data are needed to confirm our preliminary results.

The present study showed that 30.8% and 4.1% of participants experienced COVID-19 before and during Ramadan, respectively. This result could be related to the high percentage of non-vaccination (96.5%). It is known that “severe acute respiratory syndrome coronavirus 2” (SARS-CoV-2) infections can cause long term effects of COVID-19, or long-COVID when they last longer than 12 weeks [61,62]. Long-term COVID or post-COVID syndrome represents the persistence of symptoms in patients who recovered from SARS-CoV-2 infection [61]. Long-term COVID symptoms include fatigue, dyspnoea, post-activity polypnea, cough, chest pain/discomfort, resting tachycardia, palpitations, general pain, reduced pulmonary diffusing capacity, pulmonary fibrosis, sputum, stroke, arterial hypertension, myocarditis, arrhythmia, loss of smell and taste, attention disorder, hair loss, and headache [63,64]. Considering this, specific consideration is required for long-term COVID patients who want to fast and are likely to engage in PA regimens, which may increase risk of further COVID-19 complications. Long-term COVID patients are recommended to undertake precautionary measures, including medical history and physical examination before making a fasting decision, therefore, feasible guidelines may be needed for long-term COVID fasting patients.

### Strengths and Limitations

The present study has two strengths. The current study is the first to address the association of RIF with PA, anxiety, well-being, and sleep-quality during the fourth wave of the COVID-19 pandemic in Iran. Second, our sample size is large. A limitation of our study is that it used online surveys, which may cause systematic bias in responses (specific individuals may not have internet access) and decrease respondents’ attention to the questions [65]. Our study includes primarily active and inactive adults, which may limit the generalizability of the findings to the general Iranian population. Additionally, the lack of data related to anxiety, well-being, and sleep-quality outcomes before Ramadan makes it impossible to distinguish the effects of COVID-19 restrictions from RIF accurately. Finally, we did not record the number of fasted days for individuals who fasted a part of Ramadan. Further trials are therefore warranted to explore the effectiveness of partial RIF. Despite these limitations, we firmly believe that our survey study provides new information about the association of RIF with PA and the other measured health-related factors during the fourth COVID-19 outbreak in Iran. Our data might have important implications for promoting safe PA in the community by the institutions in charge during RIF amid the COVID-19 pandemic. Future studies are needed to compare similar PA-related outcomes between Ramadan and other months when fasting is not a part of religious practice.

## 5. Conclusions

Our findings appear to have important implications for all individuals who practice RIF. Given that RIF along with PA was associated with improved mental health and sleep-quality outcomes, RIF practitioners might consider PA participation during Ramadan. Similarly, policymakers may consider advocating making PA accessible, safe and inexpensive during Ramadan amid the COVID-19 pandemic. Ramadan for the year 2022 starts on 2 April and is very different from previous years due to the outbreak of new coronavirus variants. Ramadan 2022 represents a challenge for Muslims. Having a healthy lifestyle and participating in regular PA while RIF, and receiving a COVID-19 vaccination, are important strategies to improve immune function and general health and to return the world to a more normal status [4]. Patients who hesitate to fast should consult their physicians, and health providers must discuss and recommend ways to perform adequate PA while fasting during the pandemic with their patients.

## Figures and Tables

**Figure 1 jcm-11-02346-f001:**
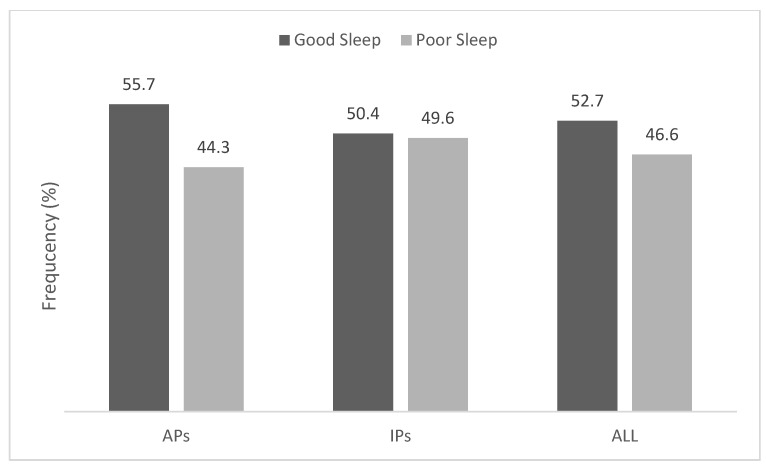
Frequencies of APs and IPs with good and poor sleep quality.

**Table 1 jcm-11-02346-t001:** Participants’ demographics by sex.

Characteristics		Men *N* (%)	Women *N* (%)	Total *N* (%)	*p*-Value
		179 (35.1)	331 (64.9)	510 (100)	
Age	(years)	31 ± 13	30 ± 12	31 ± 12	0.25
Height	(cm)	176 ± 7	163 ± 6	168 ± 9	<0.001
Weight	(kg)	79 ± 14	63 ± 13	69 ± 15	<0.001
Marital status	Single	111 (62.0)	205 (61.9)	316 (62.0)	0.99
	Married	68 (38.0)	126 (38.1)	194 (38.0)	
Schooling-level	High school	5 (2.8)	18 (5.4)	23 (4.5)	0.42
	Diploma	48 (26.8)	70 (21.1)	118 (23.1)	
	Associate	16 (8.9)	26 (7.9)	42 (8.2)	
	Bachelor	63 (35.2)	126 (38.1)	189 (37.1)	
	Master	37 (20.7)	64 (19.3)	101 (19.8)	
	Doctoral	10 (5.6)	27 (8.2)	37 (7.3)	
Employment status (pre-Ramadan)	Full-time	79 (44.1)	63 (19.0)	142 (27.8)	<0.001
	Part-time	34 (19.0)	64 (19.3)	98 (19.2)	
	Unemployed	55 (30.7)	124 (37.5)	179 (35.1)	
	Homemaker	0 (0.0)	64 (19.3)	64 (12.5)	
	Retired	11 (6.1)	15 (4.5)	26 (5.1)	
	Unable to work	0 (0.0)	1 (0.3)	1 (0.2)	
Employment status (in Ramadan)	No change	141 (78.8)	267 (80.7)	408 (80.0)	0.067
	Decreased hours	19 (10.6)	30 (9.1)	49 (9.6)	
	Remote work	3 (1.7)	19 (5.7)	22 (4.3)	
	Laid off	4 (2.2)	6 (1.8)	10 (2.0)	
	Employed	3 (1.7)	4 (1.2)	7 (1.4)	
	Increased hours	9 (5.0)	5 (1.5)	14 (2.7)	
Environment	Urban	157 (87.7)	309 (93.4)	466 (91.4)	0.091
	Suburban	8 (4.5)	9 (2.7)	17 (3.3)	
	Rural	14 (7.8)	13 (3.9)	27 (5.3)	
Location	Central	73 (40.8)	116 (35.0)	189 (37.1)	0.35
	Northern	29 (16.2)	55 (16.6)	84 (16.5)	
	Southern	19 (10.6)	29 (8.8)	48 (9.4)	
	Eastern	28 (15.6)	51 (15.4)	79 (15.5)	
	Western	30 (16.8)	80 (24.2)	110 (21.6)	
Fasting status	Full	91 (50.8)	115 (34.7)	206 (40.4)	<0.001
	Not at all	58 (32.4)	114 (34.4)	172 (33.7)	
	Partial	30 (16.8)	102 (30.8)	132 (25.9)	
Reason for not fasting	Fear of COVID-19	5 (2.8)	14 (4.2)	19 (3.7)	<0.001
	Diabetes	4 (2.2)	5 (1.5)	9 (1.8)	
	CVD	1 (0.6)	4 (1.2)	5 (1.0)	
	Kidney disease	5 (2.8)	7 (2.1)	12 (2.4)	
	Mental disease	0 (0.0)	10 (3.0)	10 (2.0)	
	Other diseases	10 (5.6)	59 (17.8)	69 (13.5)	
	Personal reasons	61 (34.1)	119 (36.0)	180 (35.3)	
Religion	Shia	153 (85.5)	304 (91.8)	457 (89.6)	0.011
	Sunni	14 (7.8)	21 (6.3)	35 (6.9)	
	Zoroastrianism	0 (0.0)	1 (0.3)	1 (0.2)	
	Other	1 (0.6)	2 (0.6)	3 (0.6)	
	Irreligion	11 (6.1)	3 (0.9)	14 (2.7)	
Ethnicity	Azari	31 (17.3)	53 (16.0)	84 (16.5)	0.2
	Kurdish	18 (10.1)	37 (11.2)	55 (10.8)	
	Turkmen	0 (0.0)	2 (0.6)	2 (0.4)	
	Lor	11 (6.1)	24 (7.3)	35 (6.9)	
	Arab	4 (2.2)	0 (0.0)	4 (0.8)	
	Baloch	2 (1.1)	3 (0.9)	5 (1.0)	
	Fars	100 (55.9)	194 (58.6)	294 (57.6)	
	Others	13 (7.3)	18 (5.4)	31 (6.1)	
Vaccination status	Yes	3 (1.7)	15 (4.5)	18 (3.5)	0.095
	No	176 (98.3)	316 (95.5)	492 (96.5)	
COVID-19 experience	Pre-Ramadan	57 (31.8)	100 (30.2)	157 (30.8)	0.53
	In Ramadan	5 (2.8)	16 (4.8)	21 (4.1)	
	Never	117 (65.4)	215 (65.0)	332 (65.1)	

COVID: Coronavirus disease. CVD: cardiovascular disease. *N*: number. Data were number (%) and mean ± standard deviation for the categorical and quantitative variables, respectively. *p*-Values represent the difference between men and women.

**Table 2 jcm-11-02346-t002:** Physical-activity (PA) behavior during Ramadan.

Characteristics		IPs *N* (%)	APs *N* (%)	Total *N* (%)	*p*-Value
		282 (55.3)	228 (44.7)	510 (100)	
Godin Leisure Score	(absolute value)	7 ± 8	127 ± 369	61 ± 254	<0.001
Strenuous PA	(min/wk)	3 ± 14	65 ± 223	31 ± 152	<0.001
Moderate PA	(min/wk)	8 ± 24	71 ± 142	36 ± 101	<0.001
Mild/light PA	(min/wk)	25 ± 170	59 ± 109	40 ± 147	0.007
Changes in PA duration	Less	76 (27.0)	96 (42.1)	172 (33.7)	<0.001
	About the same	66 (23.4)	96 (42.1)	162 (31.8)	
	More	7 (2.5)	18 (7.9)	25 (4.9)	
Common PA type	Weight training	4 (1.4)	13 (5.7)	17 (3.3)	<0.001
	Biking/cycling	5 (1.8)	21 (9.2)	26 (5.1)	
	Walking	88 (31.2)	114 (50.0)	202 (39.6)	
	Running	2 (0.7)	14 (6.1)	16 (3.1)	
	Martial arts	1 (0.4)	3 (1.3)	4 (0.8)	
	Online training	8 (2.8)	9 (3.9)	17 (3.3)	
	Others	25 (8.9)	34 (14.9)	59 (11.6)	
Changes in PA type	Very similar	74 (26.2)	118 (51.8)	192 (37.6)	<0.001
	Somewhat similar	33 (11.7)	66 (28.9)	99 (19.4)	
	Not so similar	29 (10.3)	24 (10.5)	53 (10.4)	
Common PA location	Outdoors	79 (28.0)	127 (55.7)	206 (40.4)	<0.001
	Indoors	3 (1.1)	15 (6.6)	18 (3.5)	
	In the house	52 (18.4)	65 (28.5)	117 (22.9)	
Changes in PA location	Yes	37 (13.1)	55 (24.1)	92 (18.0)	<0.001
	No	100 (35.5)	154 (67.5)	254 (49.8)	

APs: active participants. IPs: inactive participants. *N*: number. Data were number (%) and mean ± standard deviation for the categorical and quantitative variables, respectively. *p*-Values represent the difference between IPs and APs.

**Table 3 jcm-11-02346-t003:** Anxiety, well-being, and sleep-quality outcomes between IPs and APs.

Factors	IPs *N* (%) 282 (55.3)	APs *N* (%) 228 (44.7)	Total *N* (%) 510 (100)	*p*-Value
Emotional well-being	11.24 ± 3.91	12.11 ± 3.90	11.63 ± 3.93	0.014
Psychological well-being	18.11 ± 7.13	20.25 ± 6.50	19.07 ± 6.93	<0.001
Social well-being	18.05 ± 6.78	20.48 ± 6.43	19.14 ± 6.73	<0.001
Overall well-being	47.40 ± 16.22	52.84 ± 15.04	49.84 ± 15.92	<0.001
General Anxiety Disorder-7	5.77 ± 4.58	5.72 ± 4.63	5.75 ± 4.60	0.90
Pittsburgh Sleep-quality Index	6.00 ± 3.26	5.45 ± 2.87	5.75 ± 3.10	0.046

APs: active participants. IPs: inactive participants. *N*: number. Data were mean ± standard deviation. All data were analyzed using *t*-tests.

**Table 4 jcm-11-02346-t004:** Anxiety, well-being and sleep-quality outcomes based on fasting status.

	IPs (*N* = 282)				APs (*N* = 228)			
Factors	Full-Fast (*N* = 116)	Partial-Fast (*N* = 78)	No-Fast (*N* = 88)	*p*-Value	Full-Fast (*N* = 90)	Partial-Fast (*N* = 54)	No-Fast (*N* = 84)	*p*-Value
Emotional well-being	11.98 ± 3.61 ^#^	11.24 ± 4.03	10.27 ± 4.02	0.008	12.90 ± 3.69 ^#^	12.09 ± 3.31	11.26 ± 4.32	0.021
Psychological well-being	19.84 ± 7.10 ^#^	17.58 ± 7.14	16.31 ± 6.71	0.001	21.67 ± 6.43 ^#^	19.54 ± 5.26	19.20 ± 7.07	0.028
Social well-being	18.94 ± 6.33	17.50 ± 6.95	17.36 ± 7.14	0.18	21.86 ± 6.18	19.22 ± 5.69	19.82 ± 6.91	0.028
Overall well-being	50.76 ± 15.50 ^#^	46.32 ± 16.95	43.94 ± 15.79	0.009	56.42 ± 14.32 ^#^	50.85 ± 12.83	50.29 ± 16.46	0.014
General Anxiety Disorder-7	4.76 ± 4.34 ^#^	5.94 ± 4.26	6.97 ± 4.90	0.003	4.41 ± 4.08 ^#^	5.70 ± 3.76	7.13 ± 5.27	<0.001
Pittsburgh Sleep-quality Index	5.94 ± 3.35	6.32 ± 3.34	5.80 ± 3.08	0.57	4.81 ± 2.64 *	6.33 ± 2.79	5.55 ± 3.02	0.008

APs: active participants. IPs: inactive participants. *N*: number. Data were mean ± standard deviation. All data were analyzed using one-way ANOVA followed by Bonferroni’s post hoc test. * Significant difference compared with the partial fast state; **^#^** Significant difference compared with the no fast state.

## Data Availability

The data presented in this study are available on request from the corresponding author.

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
