# Peer review of "Association of Ramadan Participation with Psychological Parameters: A Cross-Sectional Study during the COVID-19 Pandemic in Iran"

_jcm, 2022, doi:10.3390/jcm11092346_

Round 1
Reviewer 1 Report
Review “Association of Ramadan Participation with Psychological Parameters: A Cross-Sectional Study During the COVID-19 Pandemic in Iran”
The paper focuses on the analysis and evaluation of COVID-19 restrictions and intermittent fasting of Ramadan during the fourth wave of COVID-19 associated with scarce results of PA, anxiety, mental health (or well-being) and quality of the sleep through an online questionnaire administered to 510 people in Iran.
The following papers are recommended for an in-depth study:
- Comparison of knowledge, attitude, socioeconomic burden, and mental health disorders of COVID-19 pandemic between general population and health care workers in Egypt. The Egyptian journal of neurology, psychiatry and neurosurgery, 57(1), 1-11.
- The effect of motor development in adolescence on cognition: A cumulative literature review, Journal of Human Sport and Exercise, 16 (2proc), 663–670. https://doi.org/10.14198/jhse.2021.16.Proc2.51
- Is the manuscript clear, relevant to the field, and presented in a well-structured way?
The paper is of interest to the scientific community, as it empirically investigates the impacts of Ramadan fasting on the psychological health Muslims the COVID-19 pandemic. What emerged is useful for expanding knowledge regarding the association between Ramadan fasting and some psycho-physical parameters, such as physical activity, mental health and sleep quality during COVID-19.
The paper is clear, and the structure is adequate: the abstract summarizes in detail the purpose of the research; the introduction is shortened, so it is advisable to expand it; the paragraph “population and methods” – and the sub-paragraphs “participants”, “measures”, “demographic information”, “main results and measures: physical activity behaviour”, “secondary results and measures: anxiety, well-being and sleep quality” and “Statistical analysis” – defines the sample, also providing information on other demographic characteristics (age, sex, height, weight, marital status, level of education, employment status before and during Ramadan and the living environment); the tests administered are valid. The next paragraph of the results accurately describes the statistical analysis conducted, and the tables provide a simplified and evident view of the procedures adopted. The discussions propose a reflection on the results by analysing the sample responses in each item. The paragraph “strengths and limitations” and the conclusions elaborate a possible application of the results with a look at future research and define the limitations of the study.
- Are the references cited current (mostly in the last 5 years)? Does it include an abnormal number of self-citations?
The number of references is adequate in consideration of the fact that the paper has the nature of empirical research focused on the analysis of the results deriving from the administration of an online questionnaire. It is also noted that most of the references are of recent years and international.
- Is the manuscript scientifically valid and is the experimental design appropriate to test the hypothesis?
The authors conduct a survey through an online questionnaire that allows to collect the behaviours of the participants, and the answers are statistically analysed in an appropriate way, supporting the results.
- Are the manuscript results reproducible based on the details provided in the methods section?
The sections “main results and measures: BP behaviour”, “results and secondary measures: anxiety, well-being and sleep quality” and “statistical analysis” clarify the selection of the sample and the tests adopted, verifying their reliability; therefore, they allow the replicability of the study.
- Are the figures / tables / images / schemes appropriate? Do they show the data correctly? Are they easy to interpret and understand? Are the data interpreted appropriately and consistently throughout the manuscript? Please include details of statistical analysis or data acquired from specific databases.
The tables clearly and correctly provide the results of the statistical analyses. Specifically, Cronbach’s alpha was calculated using the Persian version of the Godin-Shephard Leisure-Time Exercise Questionnaire (GSLTEQ), the Persian version of the Mental Health Continuum-Short Form (MHC-SF) and the Persian version of the Pittsburgh Sleep Quality Index (PSQI). In addition, Independent T-tests, test of the who square, analysis of variance (ANOVA) and Bonferroni post hoc tests were used to conduct comparative analyses.
- Are the conclusions consistent with the evidence and arguments presented?
The data analysis was carried out wisely and was interpreted correctly. It was found that there is an association between Ramadan fasting and physical activity, anxiety, well-being and results on sleep quality during the fourth wave of COVID-19 in Iran. These findings suggest that healthy adults could draw benefit from execution of physical activities while fasting, which provides support psychological independently from the they level of physical activity. However, these benefits are more significant in individuals who are physically active.
- Please review ethical statements and data availability statements to ensure they are adequate
I respect the conflict of interests; the authors declare absence of conflict of interest.
Instead, I respect the declaration on the availability of the data, data introduce yourself in this study I’m available on request of the author corresponding.
Author Response
RESPONSE TO REVIEWER 1 COMMENTS
COMMENT N°1. The paper focuses on the analysis and evaluation of COVID-19 restrictions and intermittent fasting of Ramadan during the fourth wave of COVID-19 associated with scarce results of PA, anxiety, mental health (or well-being) and quality of the sleep through an online questionnaire administered to 510 people in Iran.
The following papers are recommended for an in-depth study:
- Comparison of knowledge, attitude, socioeconomic burden, and mental health disorders of COVID-19 pandemic between general population and health care workers in Egypt. The Egyptian journal of neurology, psychiatry and neurosurgery, 57(1), 1-11.
- The effect of motor development in adolescence on cognition: A cumulative literature review, Journal of Human Sport and Exercise, 16 (2proc), 663–670. https://doi.org/10.14198/jhse.2021.16.Proc2.51
RESPONSE. Thank you. The first paper you suggested was added in the References’ list. However, the second paper is not related to our study’ topic.
COMMENT N°2. Is the manuscript clear, relevant to the field, and presented in a well-structured way?
The paper is of interest to the scientific community, as it empirically investigates the impacts of Ramadan fasting on the psychological health Muslims the COVID-19 pandemic. What emerged is useful for expanding knowledge regarding the association between Ramadan fasting and some psycho-physical parameters, such as physical activity, mental health and sleep quality during COVID-19.
RESPONSE. Thank you for this positive feedback. We are “happy” to read such “encouraging” comment.
COMMENT N°3. The paper is clear, and the structure is adequate: the abstract summarizes in detail the purpose of the research; the introduction is shortened, so it is advisable to expand it; the paragraph “population and methods” – and the sub-paragraphs “participants”, “measures”, “demographic information”, “main results and measures: physical activity behaviour”, “secondary results and measures: anxiety, well-being and sleep quality” and “Statistical analysis” – defines the sample, also providing information on other demographic characteristics (age, sex, height, weight, marital status, level of education, employment status before and during Ramadan and the living environment); the tests administered are valid. The next paragraph of the results accurately describes the statistical analysis conducted, and the tables provide a simplified and evident view of the procedures adopted. The discussions propose a reflection on the results by analysing the sample responses in each item. The paragraph “strengths and limitations” and the conclusions elaborate a possible application of the results with a look at future research and define the limitations of the study.
RESPONSE. Thank you for this positive feedback.
COMMENT N°4. Are the references cited current (mostly in the last 5 years)? Does it include an abnormal number of self-citations?
The number of references is adequate in consideration of the fact that the paper has the nature of empirical research focused on the analysis of the results deriving from the administration of an online questionnaire. It is also noted that most of the references are of recent years and international.
RESPONSE. Thank you for this positive feedback.
COMMENT N°5. Is the manuscript scientifically valid and is the experimental design appropriate to test the hypothesis?
The authors conduct a survey through an online questionnaire that allows to collect the behaviours of the participants, and the answers are statistically analysed in an appropriate way, supporting the results.
RESPONSE. Thank you for this positive feedback.
COMMENT N°6. Are the manuscript results reproducible based on the details provided in the methods section?
The sections “main results and measures: BP behaviour”, “results and secondary measures: anxiety, well-being and sleep quality” and “statistical analysis” clarify the selection of the sample and the tests adopted, verifying their reliability; therefore, they allow the replicability of the study.
RESPONSE. Thank you for this positive feedback.
COMMENT N°7. Are the figures / tables / images / schemes appropriate? Do they show the data correctly? Are they easy to interpret and understand? Are the data interpreted appropriately and consistently throughout the manuscript? Please include details of statistical analysis or data acquired from specific databases.
The tables clearly and correctly provide the results of the statistical analyses. Specifically, Cronbach’s alpha was calculated using the Persian version of the Godin-Shephard Leisure-Time Exercise Questionnaire (GSLTEQ), the Persian version of the Mental Health Continuum-Short Form (MHC-SF) and the Persian version of the Pittsburgh Sleep Quality Index (PSQI). In addition, Independent T-tests, test of the who square, analysis of variance (ANOVA) and Bonferroni post hoc tests were used to conduct comparative analyses.
RESPONSE. Thank you for this positive feedback.
COMMENT N°8. Are the conclusions consistent with the evidence and arguments presented?
The data analysis was carried out wisely and was interpreted correctly. It was found that there is an association between Ramadan fasting and physical activity, anxiety, well-being and results on sleep quality during the fourth wave of COVID-19 in Iran. These findings suggest that healthy adults could draw benefit from execution of physical activities while fasting, which provides support psychological independently from the they level of physical activity. However, these benefits are more significant in individuals who are physically active.
RESPONSE. Thank you for this positive feedback.
COMMENT N°9. Please review ethical statements and data availability statements to ensure they are adequate
I respect the conflict of interests; the authors declare absence of conflict of interest.
Instead, I respect the declaration on the availability of the data, data introduce yourself in this study I’m available on request of the author corresponding.
RESPONSE. Thank you for this positive feedback.

Reviewer 2 Report
My only additional concern is that the authors previously published a similar study on physical activity during COVID-19 restrictions:
"How physical activity behavior affected well-being, anxiety and sleep quality during COVID-19 restrictions in Iran. Eur Rev Med Pharmacol Sci, 2021."
The main difference between the present study and the previous study is that the present study occurs during Ramadan. However, the authors have not compared differences in the results of the two studies to justify publishing the present study.
Some very minor comments in the attached pdf file.

Author Response
RESPONSE TO REVIEWER 2 COMMENTS
COMMENT N°1. My only additional concern is that the authors previously published a similar study on physical activity during COVID-19 restrictions:
"How physical activity behavior affected well-being, anxiety and sleep quality during COVID-19 restrictions in Iran. Eur Rev Med Pharmacol Sci, 2021."
The main difference between the present study and the previous study is that the present study occurs during Ramadan. However, the authors have not compared differences in the results of the two studies to justify publishing the present study.
RESPONSE. Thank you for this pertinent remark. Ramadan is a very special event for a very large population worldwide. The combination of COVID-19 and Ramadan is a very unique situation, and considering the wide spread implications for so much of the world , we believe that this justifies a unique publication. In addition, after an “internal discussion” between the authors and in order to make the paper short and focused on only one aim, we have decided not to compare differences in the results of the two studies. Our initial strategy was to publish two papers with different aims. Second, after acceptance of the actual paper, we have planned to write a “short paper” or a “letter to editor” to compare the data out of Ramadan and during Ramadan. Third, the following sentence was added L: 382-384.
The authors are planning to compare the findings of the present study (ie; occurring during Ramadan 2021) with these of our previous study (ie, occurring a few months before the Ramadan 2021).
COMMENT N°2. Some very minor comments in the attached pdf file.
RESPONSE. We have considered all your remarks. Thank you.

Reviewer 3 Report
It is an interesting study using an on-line questionnaire to test the association of fasting and physical activity and other psychological parameters during COVID-19 restriction.
Major concerns are:
- Since physical activity was the primary outcome, a dichotomous GSLTEQ score dividing your participants into active and inactive seemed arbitrary. Wound the results be still valid when you use a 3- or 4-level category of physical activity?
- Partial-fast was also a heterogenous group in your study. It has been limited by the design of your questionnaire. Threrefore, we could not tell whether the partial-fast was close to 0 or near 100%. It is worth mentioning in the limitation section. And it is recommended to avoid any inferential statement with partial-fast. For example, in the Line 224, "The APs who fasted all of the Ramadan had better sleep quality outcomes than those who fasted part of Ramadan." It is a statistical inferential, but it is hard to explain why partial-fast had poorer sleep quality than no-fast.
- The Pittsburgh Sleep quality index traditionally used 5 as a cut-off point to detect poor sleep quality. The study population, either IP or AP, had an average of score near 5. It means that they had generally only mild sleep problems. Therefore, a percentage of poor sleep quality using a score >5 may be more meaningful to us.
Author Response
RESPONSE TO REVIEWER 3 COMMENTS
COMMENT N°1. It is an interesting study using an on-line questionnaire to test the association of fasting and physical activity and other psychological parameters during COVID-19 restriction.
Major concerns are:
- Since physical activity was the primary outcome, a dichotomous GSLTEQ score dividing your participants into active and inactive seemed arbitrary. Wound the results be still valid when you use a 3- or 4-level category of physical activity?
RESPONSE. Thank you for your suggestion. However, our classification of active and inactive was not arbitrary. The classification was based on results of previous studies. We have noted the following paragraph in the manuscript L125-138.
We used the Persian version of the Godin-Shephard Leisure-Time Exercise Questionnaire (GSLTEQ) to assess individual PA behavior over the previous week (H. Akbari et al., 2021; Farmanbar, Niknami, Lubans, & Hidarnia, 2013), which has been shown to have acceptable reliability in Iranian populations (reliability coefficient = 0.79) (Farmanbar et al., 2013). We divided individuals according to their GSLTEQ score: ≥ 24 (active), ≤ 23 (inactive) (Amireault & Godin, 2015). Similar to a previous study which assessed how the preventive measures impacted PA behavior and well-being of Canadians (Lesser & Nienhuis, 2020), the cut off for active was >150 min of moderate-vigorous PA per week while the related cut off for inactive was <149.9 min of moderate-vigorous PA per week (Warburton, Charlesworth, Ivey, Nettlefold, & Bredin, 2010). We also asked further questions to assess current PA behavior: (i) whether PA had changed since Ramadan, (ii) the type of PA most frequently exercised during Ramadan, (iii) whether the type of PA had changed since Ramadan, (iv) location of PA during Ramadan, and (v) whether the location had changed during Ramadan.
COMMENT N°2. Partial-fast was also a heterogenous group in your study. It has been limited by the design of your questionnaire. Threrefore, we could not tell whether the partial-fast was close to 0 or near 100%. It is worth mentioning in the limitation section. And it is recommended to avoid any inferential statement with partial-fast. For example, in the Line 224, "The APs who fasted all of the Ramadan had better sleep quality outcomes than those who fasted part of Ramadan." It is a statistical inferential, but it is hard to explain why partial-fast had poorer sleep quality than no-fast.
RESPONSE. Thank you. We have considered all your remarks.
First, we have added this paragraph in the discussion section L333-343.
The present study demonstrated that individuals who performed partial fast had poorer sleep quality than individuals who did not fast (non-significant). This result could be explained by the dangers of fasting if it is not done correctly which can increase stress levels and disrupt sleep. Partial fast could be considered compatible with non-Ramadan dietary approach. Non-RIF can take different forms such as time-restricted feeding/eating (Tinsley et al., 2017), alternate-day fasting (Kalam et al., 2019), and intermittent energy restriction (Davis et al., 2016). Although the compelling evidence of the benefits of intermittent fasting forms on physiological (Davis et al., 2016; Kalam et al., 2019; Tinsley et al., 2017) and psychological parameters (Berthelot et al., 2021), it is unknown whether the psycho-physical differences between fasting all Ramadan and fasting part of Ramadan. Thus, additional data are needed to confirm our preliminary results.
Second, we have added this sentence in the limitations subsection L376-378.
Finally, we did not record the number of fasted days for individuals who fasted a part of Ramadan. Further trials are therefore warranted to explore the effectiveness of partial RIF.
COMMENT N°3. The Pittsburgh Sleep quality index traditionally used 5 as a cut-off point to detect poor sleep quality. The study population, either IP or AP, had an average of score near 5. It means that they had generally only mild sleep problems. Therefore, a percentage of poor sleep quality using a score >5 may be more meaningful to us.
RESPONSE. Done as requested.
First, we have added a sentence in the methods section L159-150.
The global PSQI scores > 5 and ≤ 5 indicate poor and good sleep qualities, respectively (Buysse, Reynolds Iii, Monk, Berman, & Kupfer, 1989).
Second, we have added a figure (now figure 1) to clarify the percentage of APs and IPs with good and poor sleep-quality. The following sentence was added L214.
Figure 1 shows the frequencies of APs and IPs with good and sleep-quality.
Figure 1. Frequencies of APs and IPs with good and poor sleep quality
